# Navigating the Eco-Design Paradox: Criteria and Methods for Sustainable Eco-Innovation Assessment in Early Development Stages

Sarah Peigné [1], Helmi Ben Rejeb [1,*], Elise Monnier [2] and Peggy Zwolinski [1]

1    G-SCOP Laboratory, University Grenoble Alpes, CNRS, Grenoble INP, 38000 Grenoble, France
2    CEA-Liten, University Grenoble Alpes, 38000 Grenoble, France
*    Correspondence: helmi.benrejeb@grenoble-inp.fr

**Abstract:** This paper examines the growing importance of eco-innovation in aligning sustainability with technological development. It explores the 'eco-design paradox', which emphasizes the tension between the need for flexible design and the availability of data required for assessing environmental impacts during early innovation stages. This paradox presents a challenge: the lack of detailed data can have an impact on the ability to make sustainable optimal design decisions as designs are still fluid. The study investigates the essential sustainability aspects to be evaluated in the initial phases of innovation and explores the involvement of decision-makers and entrepreneurs in the sustainability assessment process. The paper uses a robust action research methodology to examine various eco-innovation diagnostic tools in collaboration with two French Institutes of Technology (FITs). A comparative analysis is conducted to assess these tools' efficacy in evaluating several dimensions of sustainability, including environmental, social, and economic aspects. Our investigation identifies key criteria that are crucial for early-stage sustainability assessments, such as innovation description, systemic vision, functionality, and stakeholder involvement. The findings will guide policy makers, researchers, and practitioners in the field of eco-innovation by providing a comprehensive framework for evaluating and promoting sustainable innovations in their early stages. The insights from our findings show how important expert knowledge is in the eco-innovation diagnostic process. They also demonstrate the need for a more integrated approach to eco-innovation.

**Keywords:** eco-innovation; sustainability assessment; entrepreneurial project; experimentation; sustainability skills





## 1. Introduction

Today, sustainability concerns are driving innovation toward eco-innovation. Collaborative research and development according to an eco-innovation approach is an interesting opportunity for companies because it is considered as a competitive factor for new products and services [1–3]. It also makes it easier to gain the support of current public policies, which are increasingly concerned with environmental challenges [4].

Eco-innovation is an important concept that is complementary to eco-design since it addresses the social challenges of innovation, in addition to the environment. While eco-innovation could include eco-design, for many references it is a broader notion involving a systemic vision of the service or object, which also includes, for example, change in economic model, use, end-of-life, and the inclusion of stakeholders [5,6]. The paradigm shift from innovation to eco-innovation is relatively new. Papers started to be published in the last decade of the past century, experiencing an upsurge in publications since 2009 [7–9]. Therefore, eco-innovation is of growing importance to policy makers, practitioners, and for research. Eco-innovation is an approach that has never ceased to evolve in recent years [10]. This evolution has occurred not only in terms of semantics but also in terms of scope, from products to services and organizations, and recently to the integration of business models [11]. With respect to the

European normative approach (NF X 30-600), eco-innovation is an environmentally friendly innovation [12]. According to the ISO 56000 standard, "an innovation can be a product, service, process, model, method, or any other entity or combination of entities. The concept of innovation is characterized by novelty and value" [13]. According to this standard, value can be both financial and non-financial, e.g., "revenues, savings, productivity, sustainability, satisfaction, empowerment, engagement, experience, or trust" [13]. This new definition of innovation highlights the different forms of values that can be created by an innovation, which are not only of economic but also of social value. Most importantly, integration of the whole life cycle is absent from the definition of innovation, whereas it absolutely needs to be considered when eco-innovating. An eco-innovation is defined by Tyl as: "an innovation process that is capable of developing concepts and/or solving problems with the possibility of addressing high systemic levels; takes into account the life cycle of the system considered; seeks a strong ambition of sustainability according to different axes: environmental, societal, and usage performance and allows to be deployed from the upstream phases of development of new offers (product, service, etc.)" [14]. Evaluation of the eco-innovation capacity of a product or a service requires the three sustainable development pillars to be considered right from the start of the innovation development phases. This multidimensional aspect makes it more complex to characterize and design [3,15].

Despite the abundance of sustainability assessment methods in the literature, there is a lack of clear guidelines for choosing the most appropriate method for specific cases, such as a research and development project [16]. In most cases, sustainability concerns emerge after the development stage of the product, when product/service design is almost finalized [17,18]. In such situations, it is too late to make the necessary changes to revise the sustainability performance of the process, the production materials, or the production location [17,19]. Even so, several studies have shown the importance of considering the environment as an initial constraint [18] or at least including the environment at the earliest possible stage in the innovation process [20,21], in order to consciously build high-performance eco-friendly innovative products or services. In particular, during the research and development stage, in which most of the final sustainability impacts are determined, a sustainability assessment is often limited by the lack of data and by continuous product modification [22,23]. This is known as the "eco-design paradox". Poudelet et al. [19] and Chebaeva et al. [24] described it as a divergence between product knowledge, which grows over development time, and the possible environmental improvement of the product, which decreases over development time. For example, knowledge regarding later upscaling, durability, application context, or use phase and waste management may be lacking. However, the majority of a product's environmental and sustainability impacts are determined during the early design stages. This includes decisions related to the product's concept, materials, and manufacturing processes, which ultimately shape its life cycle environmental footprint [24,25]. Therefore, from a sustainable development perspective, it could be interesting to have a sustainability diagnosis at the start of the design phase. The main challenge facing these diagnoses is that the sustainability benefits of innovation must be determined at environmental and social levels without knowledge of the detailed technical specifications of the new product or service, which exist only in the form of an idea or a proof of concept [22,23,26].

The discussion of the eco-design paradox highlights a critical gap in current sustainability practices, where the opportunity for environmental improvements diminishes as product development progresses. Although many studies emphasize the significance of incorporating sustainability considerations in the initial stages of eco-innovations [10], there is a lack of research on practical tools that can be effectively utilized during these early phases [21]. The importance of evaluating environmental impacts/costs with a life cycle mindset is expected to still increase further, with recent policies' reviews forecasting a significant reliance on environmental footprint methodologies in the near future [27]. While current regulations primarily aim to evaluate products at the commercial stage, forthcoming policies are expected to scrutinize fundraising to avoid financing eco-hazardous projects. This evolution underscores the paradigm shift towards embedding environmental

considerations from the outset, and even before at the idea stage. In France, this is the direction taken by research centres and organizations in charge of supporting innovation. For example, the French Institutes of Technology (FITs) are thematic and interdisciplinary technological research institutes set up by the French Government for industrial competitiveness. In order to achieve their mission of bringing out innovations in future economic sectors and supporting entrepreneurial ideas and start-ups, they need to consider sustainability during innovation. Therefore, it is necessary to propose and compare a range of eco-innovation methods that enable these FITs to perform a brief sustainability assessment during the initial evaluation phase.

Based on the above description of the context, the research question raised in this paper can be summarized as follows: What relevant criteria should be considered in an eco-innovation project to anticipate and evaluate its future sustainability? This paper explores how the eco-design paradox impacts the decision-making process in early innovation stages. Our paper investigates this area by examining the tensions between the need for flexibility during eco-innovation, and the availability of detailed sustainability data. The paper is divided into six sections. After the introduction, the second section presents the methodology. The third section discusses the background literature in order to look at the main criteria for assessing the sustainability of an innovation. The fourth section presents the experimentation, which corresponds to the processing and testing of various eco-innovation diagnoses for different projects. The fifth section gives an overview of the empirical findings, which are compared with the literature. Finally, the sixth section draws conclusions with a discussion of the theoretical and practical implications of this study, the research limitations, and directions for further research.

## 2. Methodology

The research methodology presented in this paper is based on action research principles as defined by Reason and Bradbury [28]. It needs to focus on solving very specific problems, and involves stakeholders in a collaborative, participatory, and iterative process. For these reasons, the research method has been conducted to ensure close collaboration with two French Institutes of Technology (FITs): one is working on solar energy and the other on microelectronics. The FITs play a significant role in fostering entrepreneurship in France by supporting entrepreneurs and innovative companies in various fields such as energy, micro-electronics, and information technology. The FITs operate several technology transfer and incubation programs, as well as providing funding and resources to start-ups and small businesses to help them commercialize new technologies and bring them to market. Indeed, collaborative networks with research institutes, agencies, and universities are essential to drive all types of eco-innovation [29]. Therefore, the FITs work with other organizations, such as business incubators, venture capital firms, and universities, to support the development of innovative technologies. The FITs select innovative start-up proposals through a rigorous process that includes initial screening, technical and business evaluations, and funding decisions. The proposals that pass the initial screening are then evaluated by a group of experts based on the technical feasibility, as well as for the business plan, the team, and the market potential. Based on these evaluations, the FITs decide which proposals to fund and support. Increasingly, the FITs need to consider the environmental and social impacts of the proposed technology or business idea, as well as its potential to contribute to the reduction in greenhouse gas emissions or the promotion of use of renewable energy sources. The objective of the action research presented in this paper is to understand the needs and concerns of the two FITs about the assessment of innovation sustainability, to develop context-specific solutions, and to continuously adjust the research process. The framework of this study is called the "Défi-Ino" project, which stands for "the challenge of innovation" within the eco-innovation context. "Défi-Ino" is a qualitative and collaborative research approach that involves a group of researchers from a French RTO (Research and Technology Organization) focusing on alternative energies, researchers from a laboratory working on eco-design and sustainability, and two consulting companies who

are experts in sustainable development topics. The main goal of Défi-Ino is to propose and compare a set of eco-innovation methods allowing the two FITs to conduct a quick sustainability diagnosis in the initial screening process (this should last less than 1 day and cost less than 1.000 €). The reason for keeping a variety of testing approaches is that the consultants who will conduct the audits may not have been involved in the study.

As Figure 1 shows, the methodology is based both on experimentation and the literature to determine which criteria are relevant for assessing sustainability in the upstream phases.

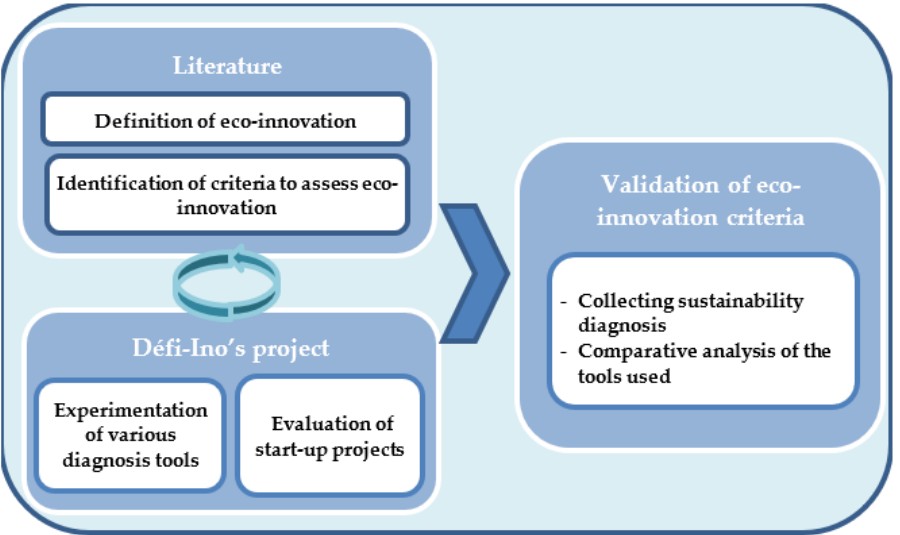

**Figure 1.** Methodology for the diagnosis of sustainability at the front-end of innovation.

Then, a test of several possible tools and methods of evaluation is conducted during an experimentation phase, in which entrepreneurial ideas during upstream phases of innovation are evaluated. To this end, the experimentation made possible the comparison of existing diagnosis practices (qualitative, quantitative, or both) focused on the environment or on all three aspects of sustainable development (economic, environmental, and societal) or on circularity. The main findings from the literature were compared with the experimentation during the "Défi-Ino" project in order to determine which criteria need to be assessed to evaluate an eco-innovation and the possibility of evaluating them in the upstream phases of a project. Finally, the relevance of the evaluation criteria is validated. From a FIT perspective, this study had two interests. First, the outcome would be useful to select the solutions for which funding is requested. Second, it can be used to empower collaborative sustainable developments during fundraising, which will help start-ups improve their solution on sustainability aspects.

## 3. Literature Review

For the state of the art, we used the following keywords: 'eco-innovation', 'tool', 'eco-design', 'diagnosis', 'sustainable innovation', 'environmental innovation', and 'eco-design paradox'. The 'Google scholar' and 'Web of Science' databases were used. Our approach was not based on a systematic review with an extensive initial pool of articles systematically narrowed down by specific exclusion criteria. Instead, our literature review was conducted in a more narrative or thematic manner, focusing on identifying and engaging with key articles that directly contribute to the theoretical and empirical foundation of our research on eco-innovation and the eco-design paradox. It appears that two recent and comprehensive literature reviews about eco-innovation and its characterization tools have already been published. Xavier et al. reviewed the current research gaps in the field of eco-innovation models [15], while Díaz-García focused on the definition of eco-innovation and determined its main drivers [7]. The literature is also mainly inspired by Tyl's thesis and paper as the framework to evaluate eco-innovative concepts [30]. Today, the life cycle analysis

(LCA) is a powerful tool for assessing environmental impacts, thanks to its standardization through the ISO 14000 standard [31]. However, it is not well-suited for the upstream phases of a project as it requires a significant amount of data and time [21,32]. Poudelet et al. (2012) found that implementation of the life cycle perspective into the product development process was facilitated by adapting LCA from a retrospective approach, quantifying the environmental impact of an existing product's life cycle, into a predictive tool to streamline design choices at an early development stage. Additionally, integrating LCA expertise into a tool to support design choices at the early development stages, sharing responsibility among different stakeholders, and relying on a specific business process reengineering methodology are all means of ensuring appropriate development for designers, as well as the successful implementation and acceptance of the tool [33]. There is currently no standard for assessing social and economic sustainability, unlike standards for evaluating quality (ISO 9001) [34], health and safety at work (ISO 45001) [35], and the environment (ISO 14001) [36]. For example, an ISO 14040-compliant LCA study for an Environmental Product Declaration or to guide a corporate technology strategy requires expert practitioners to ensure a high level of rigor and accuracy, and to interpret and communicate the results [37]. However, eco-design activities must be undertaken at the early stages when there is the greatest potential for improving environmental performance and when changes to product design are easier and cheaper to implement [38,39].

For Pialot and Millet, to assess sustainability of an eco-innovation, the environmental potentials need to be evaluated [3]. These can be either direct environmental gains generated by a change in conceptual models in design or indirect gains brought about by a transition toward a new system with a low environmental impact. Then, the viability of the dissemination of the new eco-innovative concept, with all the changes it generates, needs to be checked. A satisfactory level of viability can be measured in terms of technical feasibility, the attractiveness of the value proposition (i.e., value for the client, price, stakeholder satisfaction), and potential rebound effects. López-Forniés et al. combined these aspects into four criteria to assess eco-innovation: novelty, utility, feasibility, and environmental impacts [26]. Vallet and Tyl added the importance of using a systemic approach to assess an innovation at three main levels: the user (i.e., a description of the different users, the scenario of uses, etc.); the value chain (i.e., the different stakeholders involved in the concept across the whole value chain, and how the concept changes the initial stakeholder network); and society (i.e., the impact of the product on the value chain and society, and the dynamics of the process) [40]. Moreover, Tyl proposes a framework for the analysis of offers (products, services, etc.) qualified as eco-innovations [14]. This was built using the different characteristics and classifications found in the scientific literature. Tyl also proposed to first qualify the type of innovation (product, service, or process), followed by the environmental benefits, the ways innovation modifies the consumer's behaviour, the integration of the innovation into its context, and finally the role of institutions in the project. Building upon the previous detailed frameworks for eco-innovation assessment, there is a complementary perspective, which acknowledges the criticality of not only assessing the conceptual and environmental facets of eco-innovations but also recognizing the human capital involved. As various studies have pointed out, the transition to sustainable practices is deeply intertwined with the personal convictions of entrepreneurs and the capabilities of their teams [4,41]. This indicates the importance of evaluating the maturity level of companies, as well as the skills of their R&D and design teams, a factor that is essential for the effective implementation and realization of sustainable eco-innovations. According to Perpignan [42], this maturity could be evaluated based on the way a company considers a wide range of criteria, such as different issues of sustainability and their interactions; the ethical dimension (social responsibility, multi-criteria choice); the life cycle approach and the issue of impact transfer; the knowledge of the circular economy and eco-innovation; the integration of sustainability issues by designers; and the knowledge of the systemic dimension (scale effect, rebound effects, etc.).

Based on the literature, a set of criteria were identified to be fulfilled by an eco-innovation. As Table 1 shows, they can be grouped within four typologies: systemic vision (gathers the criteria to evaluate circularity and assess the product impacts over their entire life cycle), functionality (gathers the criteria that answer the questions 'what is it for?' and 'how will the innovation be used?'), description of the innovation (gathers the criteria to validate the novelty of the concept), and stakeholder involvement (gathers the criteria that answer the question 'what is the impact of the product on its socio-economic environment?').

**Table 1.** List of sustainability criteria of eco-innovation from the literature.

| Typology of Criteria | Criteria Observed in the Literature |
|---|---|
| Systemic | Quantify environmental impact and gains generated by the innovation [3,14,26,43] |
| | Viability of the innovation [3] |
| | Evaluate potential rebound effects [3] |
| Functionality | Assess utility of the concept [26] |
| | Feasibility of the innovation [26] |
| Innovation description | Evaluate novelty of the innovation [26] |
| | Qualify the kind of innovation [14] |
| Collaboration and stakeholder's involvement | Evaluate the integration of the innovation in its environment [14] |
| | Measure the role of institutions in the project [14] |
| | Sustainability skills of R&D and design teams [42] |

## 4. The Défi-Ino Project

### 4.1. Presentation of the Experimentation of the Eco-Innovation Diagnosis Process

The Défi-Ino project involved various stakeholders as illustrated in Figure 2. It was led by a group of project managers from two FITs. Four experts were requested to carry out a sustainability diagnosis, with the idea of deploying different methods and expertise for assessing eco-innovation, to be able to involve different consultants in the future. The FITs project managers selected these experts among their network of experts, based on their recognized expertise in sustainability assessments and their innovative approaches to eco-innovation diagnostics. The selection aimed to encompass a broad range of perspectives and methodologies, ensuring a comprehensive and multi-dimensional analysis of the eco-innovations under study. The experts' methodologies were selected for their complementariness in covering different aspects of sustainability, from environmental impacts to social implications and alignment with circular economy principles. Also, as this is an experimental project for the two FITs, tools that could be effectively applied in the early stages of innovation were prioritized to ensure relevance to the start-ups involved in the project. The preference was for tools that could be used when detailed information is limited.

Ten young companies and start-ups agreed to participate in the project. They developed innovative and technology-oriented concepts within various sectors, such as health, recycling, energy saving, and innovative systems for the building sector. Table 2 gives the list of companies and a brief description of their innovations. The primary objective of this phase in the methodology is to gather and analyse expert insights on the sustainability assessment of innovation. This involves contrasting current expert practices with the existing literature. To achieve this, two evaluations for each company's product were conducted, utilizing the respective methods of the experts. The case of company A, who developed an intelligent device to promote recycling with a connected bin, served as a dry run case for the experiment, and was thus studied by all four experts. The other companies' projects were studied by two different experts simultaneously.

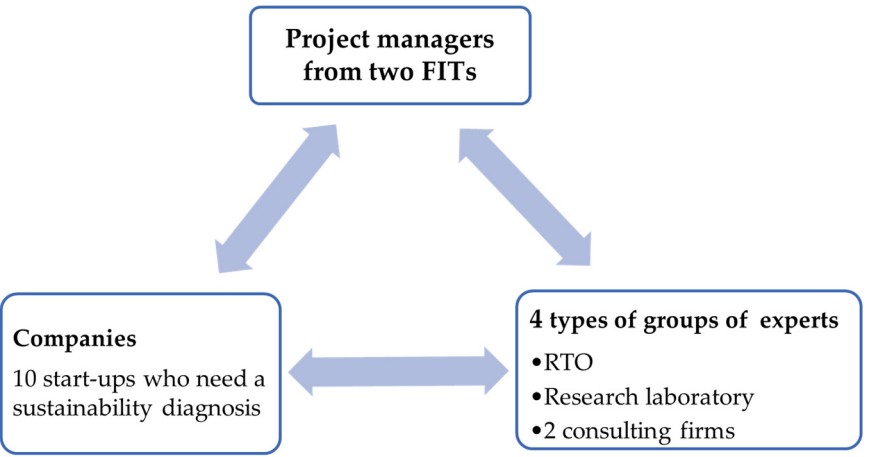

**Figure 2.** Défi-Ino experimentation stakeholders.

**Table 2.** Presentation of companies' innovation and the distribution of experts' assessment.

| Company | Innovation | Expert 1 | Expert 2 | Expert 3 | Expert 4 |
|---|---|:---:|:---:|:---:|:---:|
| A | Promote plastic recycling with connected bin | X | X | X | X |
| B | Innovative automatic doors | X | | | X |
| C | Lightweight photovoltaic panel modules | X | | X | |
| D | Construction of low-carbon buildings | X | | | X |
| E | Connected ventilation systems | X | | X | |
| F | Air quality monitoring with gas sensor technology | X | | | X |
| G | Device to collect, display and analyse odours | X | X | | |
| H | Promote glass recycling with connected bin | | X | X | |
| I | Connected prosthesis to improve patient rehabilitation | | X | | X |
| J | Card with temperature and motion sensor for smart packaging | X | X | | |

X means than the company in the row has been evaluated by the expert in the column.

In this eco-innovation diagnostic process, companies underwent a joint semi-structured interview and individual workshop conducted by each one of the two experts. Each expert employs a unique information-gathering grid during the interview, which is used to inform their individual workshop preparation. Typically, the initial interview lasts about two hours. Despite different approaches, the experts collaborate effectively without prior coordination, focusing on the company's business model, market positioning, and competitor analysis. The subsequent workshop delves deeper, examining the product life cycle, manufacturing processes, and value chain, including societal and governance aspects. Using qualitative or quantitative tools, the workshop, which can span from two hours to a full day, assesses eco-innovation criteria based on circular economy, environmental, and sustainable development principles. Following these sessions, a joint document summarizing the findings is produced. This document highlights the innovative project's strengths and areas for improvement.

### 4.2. Eco-Innovation Assessment Tools and Methods Used by the Experts

Expert 1 used the multi-impact et multi-step matrix MIME for environmental evaluation proposed by Teulon [44]. Each element of the matrix should be completed with a plus or minus sign (Table 3). A plus means that the product is harmful for the environment, while a minus means that it is not. Several plus and minus signs can be used as a means for quantification. A question mark can be added if the expert does not have enough information for the evaluation. At the end of the analysis, an overall comparison between the positive and negative impacts is conducted to see if it leans more toward positive or negative impacts. The expert also used a derived version of the MIME matrix (called MIMS) based on the same principles, which addresses the social evaluation.

**Table 3.** Example of the MIME matrix from expert 1's method.

| | Manufacturing | Packaging | Distribution | Use | End-of-Life |
|---|---|---|---|---|---|
| Energy | ++++ | +++ | − − | − − − | − |
| Air emissions | − − | ++ | + | ++ | − |
| Water emissions | ++ | + | + | ++++ | − |
| Natural resources used | − − | − | − | ++++ | − |
| Product waste | +++ | + | − | + | − |
| Toxic substances released | + | + | + | ++ | − |

The number of "+" signs indicates the level of degradation of the environmental impact category. The number of "−" signs is an indication of the level of benefit in the environmental impact category.

Expert 2 used a grid which was inspired from the APACT grid [45], used for ergonomic assessment. The used grid is a multi-impact checklist based on the comparison with an existing reference system, which analyses 90 environmental and social criteria for each life cycle phase of a product or system.

Expert 3 used the EcoAsit tool proposed by Tyl in his thesis and inspired by the ASIT tool [14]. This method is a series of affirmations to which points are attributed. These points range between 0 and 5 (5 means the affirmation is completely true, 0 means completely false). The affirmations are organized into five categories: use, responsible engagement, natural resources, natural environment, and perception.

Expert 4 used a circular canvas [46], which is a qualitative method inspired by the triple-layered business model canvas [47]. This method is composed of three steps. The first step is the functional impact, which is the value of the product. Then, the environmental impacts are listed: the benefits and the disadvantages. This consists of ten categories, including partners, users, energetic resources, and natural resources. Finally, the global viability of the product is analysed. This method concludes with a presentation of the different strategic choices to anticipate.

Expert 1's and 4's tools are similar in terms of considering a life cycle vision. However, the MIME and MIMS methods [44] resemble the LCA approach more closely, with the definitions of functional unit and qualitative impact evaluation. The methods used by experts 1, 3, and 4 could be followed by an ideation step.

The criteria analysed by the four experts are all gathered in Table 4.

**Table 4.** Criteria analysed by the method of each expert.

| Expert 1 | Expert 2 | Expert 3 | Expert 4 |
|---|---|---|---|
| • Environmental criteria considered in each phase of the life cycle:<br>  - Non-energy resources<br>  - Energy and water consumption<br>  - Presence of toxic substances<br>  - Emissions to air, water, soil<br>  - Production of waste<br>  - Others (noise, landscape, odours)<br>• Social criteria considered in each phase of the life cycle:<br>  - Governance<br>  - Social values<br>  - Employees<br>  - Local communities<br>  - Culture and society<br>  - End-user<br>  - Stakeholder scope | • Environmental criteria considered in each phase of the life cycle (emissions to water, air, land, and use of natural resources and waste generation)<br>• Extraction of materials:<br>  - Materials and substances<br>  - Consumption<br>• Manufacturing:<br>  - Packaging<br>  - Consumption<br>  - Design<br>  - Transport<br>• Use phase:<br>  - Maintenance/upkeep<br>  - Consumption<br>• Recycling:<br>  - Consumption<br>  - End-of-life<br>• Social analysis:<br>  - Workers/Employees<br>  - Local communities<br>  - End consumers<br>• Company and competition:<br>  - Market positioning<br>  - Similar product already in existence | • Use:<br>  - Meets the main need<br>  - Observed and desired use<br>  - Ease of access<br>  - Ease of use<br>• Responsible commitment:<br>  - Good working and usage conditions<br>  - Positive economic impact<br>  - Contributes to the quality of life<br>  - Respectful of differences<br>• Natural environment:<br>  - Preserves the environment<br>  - Allows and facilitates recycling<br>  - Emits little waste<br>  - Offsets emissions<br>• Natural resources:<br>  - Requires little energy<br>  - Small material and material requirements<br>  - Clean material and materials<br>  - Is sustainable<br>• Perception<br>  - Aesthetics<br>  - Sustainable design<br>  - Positive image<br>  - Prescription | • Key activities<br>  - Natural resources<br>  - Technical resources<br>  - Energy resources<br>  - Partners<br>  - Mission<br>  - Distribution<br>  - Users and contexts<br>  - Value proposition<br>  - Next use<br>• Economy<br>  - Revenues<br>  - Costs<br>• Social<br>  - Positive impacts<br>  - Negative impacts |

## 5. Results and Discussion

Our research contributes to the eco-innovation field by providing a practical exploration through the Défi-Ino project, testing and validating tools in real-world cases. Previous studies have established the foundational importance of integrating sustainability in the early stages of innovation [15,25,39]. Our work differs from theoretical models by providing empirical insights into the effectiveness of various assessment methodologies. This section presents the practical challenges and opportunities of conducting sustainability assessments while dealing with the eco-design paradox. In addressing the eco-design paradox and its implications for eco-innovation, our research underscores the important role of information in making sustainable design decisions. The challenges we identified in our study include the sources and validity of information, completeness and methodology of data collection, and availability of information. These limitations reveal key issues within the eco-innovation landscape. It is difficult to accurately assess the environmental sustainability of innovations in their early stages due to various factors. Therefore, it is important to develop reliable methods for data collection and analysis that can overcome these challenges. Establishing comprehensive metrics based on evaluation criteria for comparing eco-innovative solutions and integrating sustainability considerations into the innovation process are critical steps toward overcoming the eco-design paradox. This answers the research question regarding the relevant criteria that should be considered in an eco-innovation project to anticipate and evaluate its future sustainability.

### 5.1. Relevance and Complementarity of the Methods for a Quick Sustainability Diagnosis in the Upstream Phases

The experimentation allows different eco-innovation tools and methods to be tested to assess the same project. Some methods have already been used to assess eco-innovation, but not particularly in the upstream phases (this is the case for expert 3 and 4 methods), while others have been created or adapted for the experimentation (method 1 and 2). Despite their differences, all the experts highlighted the relevance of a diagnosis in the upstream phases of a project.

Table 5 shows the major similarities and differences between the methods. If we compare the approaches of experts 2 and 3, we can see that they are quite similar, with the difference that expert 3 does not need a reference system, unlike the multi-criteria grid of expert 2. Expert 2's method is longer and more complex than that of expert 3, as the criteria to be audited are larger than for the EcoAsit tool. Methods 1 and 2 analyse more criteria than the other tools, meaning that the diagnosis takes more than one day. The ideation process did not consider the Défi-Ino experimentation time. It was considered as a bonus following a diagnosis for those companies interested. Method 2 was created for the Défi-Ino project, so it has been tested and adapted with these companies. The same applies to the tool used by expert 1.

**Table 5.** Comparison between the methods tested.

| Criteria of Comparison | Expert 1 | Expert 2 | Expert 3 | Expert 4 |
|---|---|---|---|---|
| Comparative method | Yes | Yes | No | No |
| Qualitative/Quantitative | Qualitative | Quantitative | Qualitative | Qualitative |
| Participative method | Yes | No | No | Yes |
| Ideation | Yes | No | Yes | Yes |
| Quick diagnosis (1 day work and limited cost) | No | No | Yes | Yes |
| Maturity of the method | Quite new | New | Already tested and used | Already tested and used |

Moreover, since we are concerned with start-ups and projects at the early development phase, some technical details about the products are not available or in line with the literature dealing with the eco-design paradox [3,22–24]. This lack of data can be challenging for the quantitative-based approaches, such as expert 2's tool.

During the Défi-Ino project experimentation, it was observed that the diversity of stakeholders and varying levels of knowledge and developmental stages of participating companies significantly influenced the outcomes of sustainability diagnoses [15,48]. The evaluated projects predominantly showcased a clear and positive impact on environmental, human, or societal dimensions. The effectiveness of diagnostic tools on projects that did not initially consider sustainability aspects is a question worth exploring. The findings suggest that the diagnostic process holds the most significance for products and companies in the early stages of eco-innovation maturity [1,49]. To sustain the momentum of eco-innovation actions for such entities, it is highly recommended to combine diagnostic assessment, ideation sessions, and subsequent follow-up meetings a few months later. This approach ensures that the initial insights and recommendations from the diagnosis are continuously integrated into the company's innovation process, fostering a culture of sustainability. Conversely, if a company already has a high degree of eco-innovation maturity and awareness, the diagnostic process will reaffirm their existing strategies and practices. In these cases, the company's advanced understanding of eco-innovation principles allows them to derive similar benefits from the diagnosis as those less familiar with sustainability considerations. The

diagnostic tools and methodologies are adaptable to a wide range of eco-innovation maturity levels within the start-up ecosystem. This distinction emphasizes their adaptive nature.

### 5.2. Relevant Criteria to Be Evaluated at the Upstream Stages of Eco-Innovation

This section compares the criteria identified in the literature with those used in the experimentation. It highlights the criteria that need to be considered in the early stages of a project to ensure its future sustainability.

First, as seen in Table 6, all the criteria observed in the literature are taken into account during the experimentation phases in the eco-innovation tools tested. Quantification of the environmental gains or impacts was difficult in the early development stage as it is full of uncertainty. For example, all the production processes and materials had not yet been chosen. Also, the consumption of energy, raw materials, and resources at commercial scale were unknown. Assessing the ways innovation modifies the consumer's behaviour and assessing utility of the concept were also important key points in the diagnosis delivered by the experts. All the diagnosis tools highlight whether the concept meets a consumer's need. The study illustrates varying approaches in evaluating companies' knowledge and skills regarding sustainability and environmental impacts. All four methods emphasize the importance of a company's awareness for sustainability issues. However, their assessment techniques and focus areas differ. Methods 1 and 2 stand out for their thorough evaluation of environmental and social impacts across each life cycle phase, indicating a comprehensive approach to assessing a company's depth of understanding in these areas. Method 2 stands out by estimating certain quantitative values using available data. In contrast, method 3 and method 4 appear to have a more focused approach. Method 3 primarily concentrates on the social responsibility aspect of sustainability, shedding light on the company's ethical and social awareness. Method 4, on the other hand, prompts companies to consider local suppliers, indirectly assessing their understanding of sustainability's systemic dimensions. The study reveals that while CEOs generally show a high level of confidence and understanding of sustainability, the depth and accuracy of this knowledge vary. Some companies might focus narrowly on certain aspects, like social impact, without fully addressing broader environmental concerns or potential rebound effects. This suggests a need for tools that can effectively discern between genuine comprehensive eco-innovation approaches and more superficial conceptually virtuous ones. The diverse assessment methods employed by each tool, therefore, play a crucial role in accurately gauging a company's real understanding and implementation of sustainability and environmental impact measures.

**Table 6.** Criteria from the literature compared to the experimentation.

| Typology of Criteria | Criteria Observed in the Literature | Expert 1 | Expert 2 | Expert 3 | Expert 4 | Degree of Difficulty to Assess at the Early Innovation Stage |
|---|---|---|---|---|---|---|
| Systemic | Quantify environmental impact and gains generated by the innovation [3,14,26,43] | x | x | x | x | Hard, because the companies do not have all the answers yet (which materials, where to produce, how much can be sold, etc.) |
| | Viability of the innovation [3] | x | | x | | |
| | Evaluate potential rebound effects [3] | x | x | | | |
| Functionality | Assess utility of the concept [26] | | x | x | x | Easy, because only the concept needs to be defined for this item |
| | Feasibility of the innovation [26] | x | x | x | | |

**Table 6.** *Cont.*

| Typology of Criteria | Criteria Observed in the Literature | Expert 1 | Expert 2 | Expert 3 | Expert 4 | Degree of Difficulty to Assess at the Early Innovation Stage |
|---|---|---|---|---|---|---|
| Innovation description | Evaluate novelty of the innovation [26] | | x | x | | Easy. It is usually the first thing explained by the companies. |
| | Qualify the kind of innovation [14] | | x | x | x | |
| Collaboration and stakeholder's involvement | Evaluate the integration of the innovation in its environment [14] | x | x | x | x | Medium. Depends on the maturity of the company releasing the new concept. A very new company has a lot of uncertainty without always knowing the production means or place. |
| | Measure the role of institutions in the project [14] | x | x | x | x | |
| | Sustainability skills of R&D and design teams [42] | x | x | | | |

The four methods prove complementarity in evaluating a company's awareness of sustainability issues, with all methods indicating a life cycle approach and consideration of social, economic, and environmental impacts. Each tool rates companies on their understanding of sustainability, life cycle approach, knowledge of circular economy and eco-innovation, and integration of sustainability issues by designers. While all tools are effective in some aspects, they vary in their assessment of a company's ethical dimensions and systemic knowledge. This study underscores the value of early-stage diagnostics in raising start-up managers' sustainability awareness and suggests the addition of an ideation process post-diagnosis to further guide start-ups towards sustainability. The findings confirm previous findings in the literature for ideation as a method to enhance eco-innovation, emphasizing its role in stimulating creativity and ensuring that sustainability extends beyond a mere technical perspective [3,11,50]. The systemic approach in the Défi-Ino experimentation is treated by reviewing all the life cycle phases of the innovation to gain an overall vision. The project did not show any new criteria or specific elements that needed to be taken into account compared to the literature. However, it did highlight that, at the early design stage, companies do not have all the answers to the experts' questions. This is, therefore, the ideal time to evolve the concept and guide the company toward the broad direction of sustainability [51,52]. The amount of data, even in this limited phase of the project, has allowed the experts to identify the sensitive and beneficial points for the project to be sustainable. Recognizing the early stage of start-ups as a prime opportunity for impactful guidance [7,29], our approach to eco-innovation diagnostics is designed to steer these emerging companies towards sustainability, embedding it into their strategic foundation from the outset. This was the case, for example, for project I for the design of connected prosthesis to improve patient rehabilitation. The diagnostics highlighted the need to adopt technological sobriety in production to avoid unnecessary electronic components. Furthermore, they suggest that electronic kits be designed for reusability, reparability, and eventually recyclability across a broader patient base, thereby compensating for their environmental impact. For project F (a gas sensor for air quality monitoring), a careful attention was recommended to the scale effects (in scenarios targeting 1 million instruments) and rebound effects (where the positive impact of reducing food waste is balanced against the impacts of instrument production and digital usage). This underscores the need for comprehensive life cycle impact assessments during the development of projects [33]. In addition to the eco-innovation guidance, the Défi-Ino experiment shows that, from the early stage of project development, it is possible for FITs to screen projects to finance those that seem most economically, socially, and environmentally sustainable. Nevertheless, the auditor's knowledge and skills are crucial in accurately and effectively

assessing eco-innovation, as highlighted in the literature [2,39,53]. Therefore, it is important to have strict criteria when selecting experts for the eco-innovation diagnostic process, whatever the method is for assessing eco-innovation.

## 6. Conclusions

One of the objectives of this research is to determine the relevant criteria to be evaluated from the upstream phases of a project to anticipate its future sustainability. A literature review was conducted, and "Défi-Ino" experimentation took place, where several projects were audited using different methods, but following the same approach: preparation, workshop, and the delivery of a diagnosis of sustainability for the innovation. The main challenge in such studies with the analysis of early-stage products is the lack of data on the entire life cycle and on the three dimensions of sustainable development, a gap also highlighted in many previous papers and is referred as the eco-design paradox.

This study showed that none of the methods could fully evaluate all the criteria, but all were able to highlight sustainability hotspots. To answer the research question "what criteria should be considered in an eco-innovation project to anticipate/guarantee its future sustainability?" at the front-end of innovation, the methodology presented in this study, based on the literature and the comparative study of four experts' evaluation methods, identified the most important criteria to validate at the beginning of the entrepreneurial project. In order to anticipate impacts of an entrepreneurial project, a quick diagnosis must integrate the following main criteria: innovation description, systemic vision, functionality, and stakeholder involvement.

Analysis of this experience has allowed us to make the following recommendations. The diagnosis is dependent on the person who performs it. Thus, to be efficient and impactful, the auditor needs to be an expert in sustainability issues and needs to master his tools. Moreover, the majority of eco-innovation tools have been developed for use by environmental experts, while the start-up environment implies a high managerial turnover. This means that diagnoses will no longer be valid, and that managers' awareness will probably be lost when their start-up is acquired by another company, which is a frequent occurrence. It also seems important to question the technological maturity of the product and the maturity of the company with regard to eco-innovation for better efficiency of the diagnosis. The experimentation shows that eco-innovation evaluation in the early phase of project development would be more useful for start-ups that are not familiar with sustainability challenges. Thanks to the follow-up of start-ups after the experimentation, each company has understood the benefits of sustainability in their innovation and continues to work in its own way. For example, in one company, an employee now devotes one day a week to pursuing the eco-design approach, while another employee is searching for solutions for a specific topic raised in the diagnosis (recycling at end-of-life). All companies in the experimentation have understood the need to have a life cycle thinking and systemic approach in innovation.

Although the Défi-Ino project was implemented for two sectorial FITs (one dedicated to solar energy and the other one to microelectronics), the methodology demonstrates potential for scalability across various sectors and regions. The project's success in identifying sustainability hotspots in start-ups indicates its applicability beyond the initial context. The collaboration model, involving a diverse set of stakeholders from project managers to environmental experts, provides a robust framework that can be replicated in other ecosystems, fostering a broader adoption of eco-innovation practices. Nevertheless, the project's approach can be tailored to suit different sectors or countries by adjusting and adding criteria based on industry or country-specific sustainability challenges and opportunities.

While this analysis focuses on the content of a diagnosis, in further studies, it could be interesting to analyse the format of the diagnosis tool. For example, it could be interesting to determine whether a participative tool is better than a non-collaborative tool to impact the audited company. The findings of this study could support the efforts of the FITs during

a formal process for evaluating and selecting projects, where each FIT will benefit from business incubation services, technical support, and funding, to help early-stage start-ups bring their ideas to market. Sustainability validation at the upstream phases could be considered, in addition to more traditional criteria such as alignment of the project with the FIT's research priorities, the potential for impact in the relevant field, the technical feasibility of the project, and the quality of the team and the business plan.

Finally, it is acknowledged that one main driver for eco-innovation is regulation. Today at the early development stage, a sustainable diagnosis to evaluate an eco-innovation is not mandatory. In further studies, it would be interesting to check start-ups' motivation in conducting such a diagnosis.

**Author Contributions:** Methodology, S.P., H.B.R., E.M. and P.Z.; conceptualization, S.P., H.B.R. and P.Z.; data curation, S.P.; formal analysis, S.P.; writing—original draft, S.P. and H.B.R.; writing—review and editing, H.B.R., E.M. and P.Z. All authors have read and agreed to the published version of the manuscript.

**Funding:** This study is part by the French National Program «IRT Nanoelec» under Grant ANR-10-AIRT-05. It was supported by two French Institutes of Technology (FITs): the energy transition institute ITE INES.2S (focusing on the solar energy), and the technological and research institute IRT Nanoelec (focusing the microelectronic sector).

**Data Availability Statement:** The data presented in this study are available on request from the corresponding author. The data are not publicly available due to confidentiality reasons.

**Acknowledgments:** The authors acknowledge the collaboration with all the experts involved in the study and the SMEs managers involved in the Défi-Ino project for their participation in interviews and surveys.

**Conflicts of Interest:** The authors declare no conflicts of interest.

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
