# Peer review of "Navigating the Eco-Design Paradox: Criteria and Methods for Sustainable Eco-Innovation Assessment in Early Development Stages"

_sustainability, doi:10.3390/su16052071_

Round 1

Reviewer 1 Report

Comments and Suggestions for Authors

Dear authors,

I am pleased to have the opportunity to review this research paper. Its effort to explore the critical area of eco-innovation, particularly within the framework of the Défi-Ino project, represents a valuable contribution to our understanding of sustainability practices in the business sector. However, while the study shows a clear commitment to advancing eco-innovation, there are several aspects, detailed below, which, if improved, could significantly enhance the clarity, impact, and applicability of this investigation.

·         The introduction section could be strengthened by clearly establishing the specific gaps in the current literature that this study aims to address. Moreover, adding more recent studies that underline the growing importance of eco-innovation in sustainability efforts could provide a more compelling rationale for the study.

·         Consider integrating more recent literature in the literature review section to ensure that the review reflects the latest developments and debates in the field, highlighting conflicting perspectives or emerging trends to enrich the analysis and demonstrate a critical engagement with the subject matter.

·         Regarding to Défi-Ino Project section, may be enhanced by detailing the criteria for selecting the specific eco-innovation diagnostic tools tested, explaining the rationale behind the choice of tools and how they complement each other.

·         While the results outline the effectiveness of different eco-innovation assessment tools, this section could benefit from discussing the results in the context of existing literature, a deeper analysis of the implications, and highlighting what is genuinely novel about this research.

·         Consider offering specific, actionable policy recommendations based on the research outcomes in the conclusion section, comment the potential scalability of the Défi-Ino project and its applicability to other contexts, further develop recommendations for future research.

Estoy impaciente por ver los resultados de estas modificaciones, las cuales seguro que mejorarán la claridad y calidad de este excelente trabajo de investigación.

Kind regards

Comments on the Quality of English Language

Minor editing of English language required

Reviewer 2 Report

Comments and Suggestions for Authors

The thesis that the increasing importance of eco-innovation in adapting environmental sustainability to technological progress is limited by the "ecodesign paradox - authors of the article" - seen as the tension between the need for flexible design and the availability of data required to assess environmental impact at the early stages of innovation...

Reviewer 3 Report

Comments and Suggestions for Authors

Quality of Structure and Clarity of the text, but in the conclusion it´s not refered the future work of this  investigation/project/work.

Reviewer 4 Report

Comments and Suggestions for Authors

1-In the literature review (point 3) it is not clear how many total articles were found, how many were rejected, what the selection criteria were, and how many were taken into account

2-When citing the methodology in table 1, it is recommended to list it in the same order in which the table appears

3-Experts use different methods and tools, and a comparison is made, wouldn't a comparison using the same methods be more reliable? Why was this not done? It is reasonable that if you compare with different methods the areas of focus differ. Why the choice of different methods? Justify

4-In the start-up phase, companies do not have all the answers to the experts' questions. Instead, it is considered to be the ideal time to evolve and guide the company in the broad direction. It should be explained

5-In the initial phase it has allowed experts to identify sensitive and beneficial points, it would be advisable to mention some of them (line 453)

6-The conclusions do not include bibliographical citations

Reviewer 5 Report

Comments and Suggestions for Authors

This is a timely and well written paper. My only suggestion is that the introduction should be shortened. At present it is over two pages and suffers from a lack of paragraphing. Thus, I suggest reducing the introduction to just a few critical paragraphs. The rest of the paper is fine. 

Round 2

Reviewer 1 Report

Comments and Suggestions for Authors

Dear authors,

Congratulations.

Kind regards